:◯: PLOS | ONE

# Utilization of archived neonatal dried blood spots for genome-wide genotyping

**Pagna Sok** [1], **Philip J. Lupo** [1], **Melissa A. Richard**[1], **Karen R. Rabin**[1], **Erik A. Ehli** [2], **Noah A. Kallsen**[2], **Gareth E. Davies**[2], **Michael E. Scheurer**[1], **Austin L. Brown**[1]*

**1** Department of Pediatrics, Hematology-Oncology Section, Baylor College of Medicine, Houston, Texas, United States of America, **2** Avera Institute for Human Genetics, Sioux Falls, South Dakota, United States of America

* Austin.Brown@bcm.edu

**Data Availability Statement:** Data cannot be shared publicly because participants did not provide informed consent for the deposition of genetic data. Data are available from the EpiCenter at Baylor College of Medicine (contact via

## Abstract

### Introduction

Heel pricks are performed on newborns for diagnostic screenings of various pre-symptomatic metabolic and genetic diseases. Excess blood is spotted on Guthrie cards and archived by many states in biobanks for follow-up diagnoses and public health research. However, storage environment may vary across biobanks and across time within biobanks. With increased applications of DNA extracted from spots for genetic studies, identifying factors associated with genotyping success is critical to maximize DNA quality for future studies.

### Method

We evaluated 399 blood spots, which were part of a genome-wide association study of childhood leukemia risk in children with Down syndrome, archived at the Michigan Neonatal Biobank between 1992 and 2008. High quality DNA was defined as having post-quality control call rate $\geq$ 99.0% based on the Illumina GenomeStudio 2.0 GenCall algorithm after processing the samples on the Illumina Infinium Global Screening Array. Bivariate analyses and multivariable logistic regression models were applied to evaluate effects of storage environment and storage duration on DNA genotyping quality.

### Results

Both storage environment and duration were associated with sample genotyping call rates (p-values < 0.001). Sample call rates were associated with storage duration independent of storage environment (p-trend = 0.006 for DBS archived in an uncontrolled environment and p-trend = 0.002 in a controlled environment). However, 95% of the total sample had high genotyping quality with a call rate $\geq$ 95.0%, a standard threshold for acceptable sample quality in many genetic studies.

### Conclusion

Blood spot DNA quality was lower in samples archived in uncontrolled storage environments and for samples archived for longer durations. Still, regardless of storage environment or

epicenter@bcm.edu) for researchers who meet the criteria for access to confidential data.

**Funding:** This work was supported in part by funding from the Cancer Prevention and Research Institute of Texas (RP170074, MPI: Rabin/Lupo) and the National Cancer Institute (K07 CA218362, PI: Brown). The funders had no role in study design, data collection and analysis, decision to publish, or preparation of the manuscript.

**Competing interests:** The authors have declared that no competing interests exist.

duration, neonatal biobanks including the Michigan Neonatal Biobanks can provide access to large collections of spots with DNA quality acceptable for most genotyping studies.

## Introduction

Blood obtained from heel pricks of newborn infants are spotted on Guthrie cards and collected as part of newborn screening (NBS) programs to identify potentially life-threatening metabolic and genetic disorders including sickle cell disease, phenylketonuria, and numerous other conditions associated with long-term morbidity [1–7]. Many states store excess or residual dried blood spots (DBS) that are not needed for the NBS in long-term repositories. In states that permit the use of these DBS for biomedical research, DBS can then be linked to disease registries, providing a rich resource for population-based epidemiologic studies for conditions not typically evaluated as part of NBS programs [8, 9]. Thus far, conditions studied have included type 1 diabetes, multiple sclerosis, and childhood cancers among many others [10–12]. While DBS are increasingly being utilized to conduct genome-wide association studies for a range of phenotypes [13, 14], DBS may be in archive for decades under various conditions, which may affect the quality of DNA quality retrieved for research purposes [15–17].

The Michigan Neonatal Biobank (MNB) is a storage and management center for archiving DBS collected as part of the Michigan NBS program. The MNB has archived over five million DBS to date, which represents almost every live birth in the state since 1987. We examined the effects of multiple factors, including storage environment and storage duration, on the quality of DNA extracted from DBS retrieved from the MNB for genome-wide genotyping. Our overall goal was to understand the potential compromising factors to be considered when identifying DBS while still maximizing samples to utilize for genetic epidemiologic studies. Particularly, for genomic analyses of rare childhood conditions, utilizing archived DBS that are several decades old may be necessary to amass an adequate sample size to achieve statistical power for rare variants evaluation.

## Methods

Neonatal DBS were obtained from MNB as part of a genome-wide association study of childhood leukemia risk in children with Down syndrome [18], which permits all original 425 genotyped samples from children born between 1992 and 2008 to be evaluated. Birth records at the Michigan Department of Health and Human Services were linked to the records in the Michigan Cancer Surveillance Program to identify individuals with Down syndrome with and without acute lymphoblastic leukemia in the Michigan Cancer Surveillance program. This study was approved by the Committee for the Protection of Human Subjects of the Health and Human Services Agency of the State of Michigan and the Institutional Review Board of Baylor College of Medicine and did not involve the use of any personal identifiers.

### DNA extraction

DNA extraction and genotyping for all DBS samples were conducted in 2017. DNA extraction was performed using the GenTegra (Pleasanton, CA) GenSolve Whole Blood DNA recovery kit according to the manufacturer protocol, which allows for DNA recovery from Whatman FTA (Flinders Technology Associates) cards, regardless of storage time. For each unique sample, six 3-mm punches were combined into a single 1.5 mL microtube containing the specified amount of GenTegra Recovery Solution A/Proteinase K solution. Following incubation in a

heated shaker for a minimum of 1 hour, the tubes were centrifuged to collect any condensation. Remnant punches were removed before the solution was mixed with GenTegra Recovery Solution B. The entire volume was then transferred to either the NucleoSpin XS (Macherey-Nagel Inc., Bethlehem, PA) or Qiagen Micro (Hilden, Germany) spin column for DNA purification according to manufacturer recommendations. Total DNA was then eluted from each respective spin column by addition of approximately 35 μL of AE buffer. Quantification of the purity of the extracted DNA was performed using the Thermo Fisher Scientific NanoDrop 2000 Spectrophotometer (Wilmington, DE).

### DNA genotyping and quality control

Extracted DNA was genotyped at the Avera Institute for Human Genetics (Sioux Falls, SD) using the Illumina (San Diego, CA) Infinium Global Screening Array BeadChip, which captures approximately 700,000 single nucleotide polymorphisms (SNPs). These genotyped SNPs or variants were called with GenomeStudio 2.0 software (San Diego, CA). The analysis was restricted to variants on chromosomes 1 to 20 and 22. Variants on chromosome 21 (n = 9,443) were excluded because calling for trisomic variants is not currently supported in GenomeStudio 2.0. We implemented conservative quality control criteria based on the GenomeStudio clustering algorithm metrics [19] rather than criteria often used in genome-wide association studies [20, 21], to filter poor-performing probes and specifically evaluate sample call rate from DBS. In particular, variants with a call rate < 90% (n = 17,804), minor allele frequency < 1% (n = 150,292), and cluster separation < 0.3 (n = 20,468) were removed. Final call rates were calculated among the 472,019 variants that passed quality control.

### DBS call rate as primary outcome

GenomeStudio assigned variants genotyping based on a GenCall score generated through a calling algorithm. GenCall score is a quality metric between 0 and 1 that describes sample clustering, where a low score is assigned to samples that locate away from a cluster center. A low GenCall score is an indication of unreliable genotype calls, and typically, variants with a score < 0.15 are not assigned a genotype. Hence, sample call rate is the proportion of genotyped or called variants for an individual sample that were successfully determined by the GenomeStudio calling algorithm after initial removal of variants that did not pass quality control. Sample call rate was used as the primary outcome to evaluate DBS quality. Call rates were dichotomized on a stringent threshold such that a sample call rate ≥ 99.0% was classified as evidence of higher DNA quality while < 99.0% sample call rate indicated poorer DNA quality.

### DBS storage characteristics as primary exposure

At MNB, DBS collected between 1996 and 2008 are kept in a controlled environment at 70 degrees Fahrenheit and 35% humidity while older DBS collected prior to 1996 are stored in an uncontrolled temperature and controlled (35%) humidity environment. Thus, the primary independent variables evaluated as potential factors differentiating DNA quality were storage environment and storage duration.

### Statistical methods

Descriptive statistics, including counts and percentages for categorical variables and median and ranges for continuous variables, were calculated for the overall sample and compared across samples with call rate ≥ 99.0% or < 99.0%. Differences between sample quality and independent variables, including storage environment, storage duration, leukemia status,

infant sex, maternal race/ethnicity, total DNA yield, and 260/280 measurement (indicative of DNA quality) were evaluated using Pearson chi-square, Fisher's exact, or Wilcoxon rank-sum test. Since storage environment was correlated with year of sample collection, we fitted multi-variable logistic regression models to evaluate the association between period of sample collection and sample quality, stratified by storage environment: 1) collected prior to 1996 (n = 101) and 2) collected on or after 1996 (n = 298). Logistic regression models were adjusted for total DNA yield and maternal race/ethnicity. All statistical analyses were conducted in R 3.5.2 and a significance threshold p-value < 0.05 was applied. Finally, to identify if any chromosomal regions were more susceptible to DNA degradation due to differences in storage conditions, we applied Fisher's exact test to evaluate missing call counts across post-quality control geno-typed SNPs in PLINK 1.9 using storage environments as the dependent variable [22, 23]. Variants with 0% or 100% missing call frequency were not evaluated.

## Results

### DBS call rate quality

To compare sample quality of the 425 DBS collected from newborns born between 1992 and 2008, 26 samples were excluded due to differences in DNA extraction methods. Among the 399 remaining samples, 75% were collected between 1996 and 2008 and stored in a controlled environment at 70 degrees Fahrenheit and 35% humidity (Table 1). The proportion of DBS collected and archived each year constituted between 4–8% of the total for the 16-year time period, with the exception of 2% in the year 2004. The majority of the samples were collected from individuals without a known diagnosis of childhood leukemia (94%), male sex (52%), and non-Hispanic white maternal race/ethnicity (77%).

The median sample call rate was 98.8% (range = 79.9–99.0%) prior to variant filtering. Variant filtering statistically improved call rate (median = 99.9%, range = 80.4–99.9%, p-value < 0.001) where 339 (85%) DBS had a call rate ≥ 99.0% after filtering (Table 1). The number of samples with call rate < 99.0% occurred more frequently in those collected prior to 1996 that were stored in a laboratory retention center without a controlled environment (p-value < 0.001). In contrast, the proportion of samples with call rate ≥ 99.0% archived on or after 1996 exceeded 85% every year, with the exception of 1997 (79%) and 1998 (79%). Maternal race/ethnicity was significantly associated with call rate (p-value = 0.04); however, this association was no longer statistically significant in models accounting for year of birth, suggesting that differences in call rates between racial/ethnic groups was confounded by changes in demographic dynamics of the population during the DBS collection period.

### DBS call rate quality by storage environment and duration

We observed improvements in median sample call rate by year of DBS collection and archive environment (Fig 1). This trend was apparent both prior to and following variant filtering. To evaluate if the association between sample call rates and date of sample collection remained after accounting for differences in storage conditions, we conducted multivariable logistic regression stratifying on storage environment (Table 2). Independent of storage environment, there was also a positive correlation between the proportion of DBS with call rate ≥ 99.0% and more recent collection years. This was true for both DBS collected prior to 1996 (p-value for trend = 0.006) and samples collected on or after 1996 (p-value for trend = 0.002). We did not identify statistically significant associations between storage environments (S1 Table) and leukemia status (p-value = 0.9), infant sex (p-value = 0.9), maternal race/ethnicity (p-value = 0.9), or total DNA extracted (p-value = 0.05). Although DBS placement along the genotyping array was randomized, we further evaluated DBS placement as a potentially factor affecting call rate

**Table 1. Associations between residual DBS characteristics and call rate among DBS archived at the Michigan Neonatal Biobank between 1992 and 2008.**

| Residual DBS characteristics | Overall (n = 399) | Sample Call Rate | | p-val[1] |
|---|---|---|---|---|
| | | ≥ 99.0% (n = 339) | < 99.0% (n = 60) | |
| **Storage environments, n (%)** | | | | <0.001 |
| Uncontrolled environment[2] | 101 (25%) | 54 (16%) | 47 (78%) | |
| Controlled environment[3] | 298 (75%) | 285 (84%) | 13 (22%) | |
| **Year DBS collected, n (%)** | | | | <0.001 |
| 1992 | 26 (7%) | 11 (3%) | 15 (25%) | |
| 1993 | 24 (6%) | 8 (2%) | 16 (27%) | |
| 1994 | 29 (7%) | 17 (5%) | 12 (20%) | |
| 1995 | 22 (6%) | 18 (5%) | 4 (7%) | |
| 1996 | 26 (7%) | 26 (8%) | 0 (0%) | |
| 1997 | 24 (6%) | 19 (6%) | 5 (8%) | |
| 1998 | 24 (6%) | 19 (6%) | 5(8%) | |
| 1999 | 32 (8%) | 31 (9%) | 1 (2%) | |
| 2000 | 16 (4%) | 16 (5%) | 0 (0%) | |
| 2001 | 25 (6%) | 25 (7%) | 0 (0%) | |
| 2002 | 28 (7%) | 28 (8%) | 0 (0%) | |
| 2003 | 17 (4%) | 15 (4%) | 2 (3%) | |
| 2004 | 8 (2%) | 8 (2%) | 0 (0%) | |
| 2005 | 24 (6%) | 24 (7%) | 0 (0%) | |
| 2006 | 33 (8%) | 33 (10%) | 0 (0%) | |
| 2007 | 14 (4%) | 14 (4%) | 0 (0%) | |
| 2008 | 27 (7%) | 27 (8%) | 0 (0%) | |
| **Leukemia status, n (%)** | | | | 0.6 |
| Leukemia | 23 (6%) | 21 (6%) | 2 (3%) | |
| No leukemia | 376 (94%) | 318 (94%) | 58 (97%) | |
| **Infant sex, n (%)** | | | | 0.6 |
| Male | 206 (52%) | 173 (51%) | 33 (55%) | |
| Female | 193 (48%) | 166 (49%) | 27 (45%) | |
| **Maternal race/ethnicity, n (%)** | | | | 0.04 |
| Non-Hispanic White | 308 (77%) | 264 (78%) | 44 (73%) | |
| Non-Hispanic Black | 57 (14%) | 44 (13%) | 13 (22%) | |
| Non-Hispanic Asian | 13 (3%) | 10 (3%) | 3 (5%) | |
| Hispanic | 21 (5%) | 21 (6%) | 0 (0%) | |
| **Total DNA yield in μg, median (range)** | 5.3 (2.3–14.4) | 5.3 (2.4–14.4) | 4.8 (2.3–13.3) | 0.3 |

[1]Pearson chi-square, Fisher's exact, or Wilcoxon rank-sum p-value comparing residual DBS characteristics by sample call rate.

[2]Uncontrolled temperature, 35% humidity.

[3]70 degrees Fahrenheit, 35% humidity.

and did not observed any significant associations with call rate, storage environment, or DBS collection year (S2 Table). Moreover, DBS placement was not statistically associated with call rate (p-value = 0.5, data not shown) in the multivariable models accounting for total DNA yield, DBS collection year, maternal race/ethnicity, and storage environment. Therefore, DBS placement was not included in the final models presented.

Finally, we evaluated SNP missing frequency by storage environment and found random missing frequency across all chromosomes (Table 3, S1 Fig). Genome-wide statistically significant (p-value < 5×10$^{-8}$) differential missingness was observed between storage environments

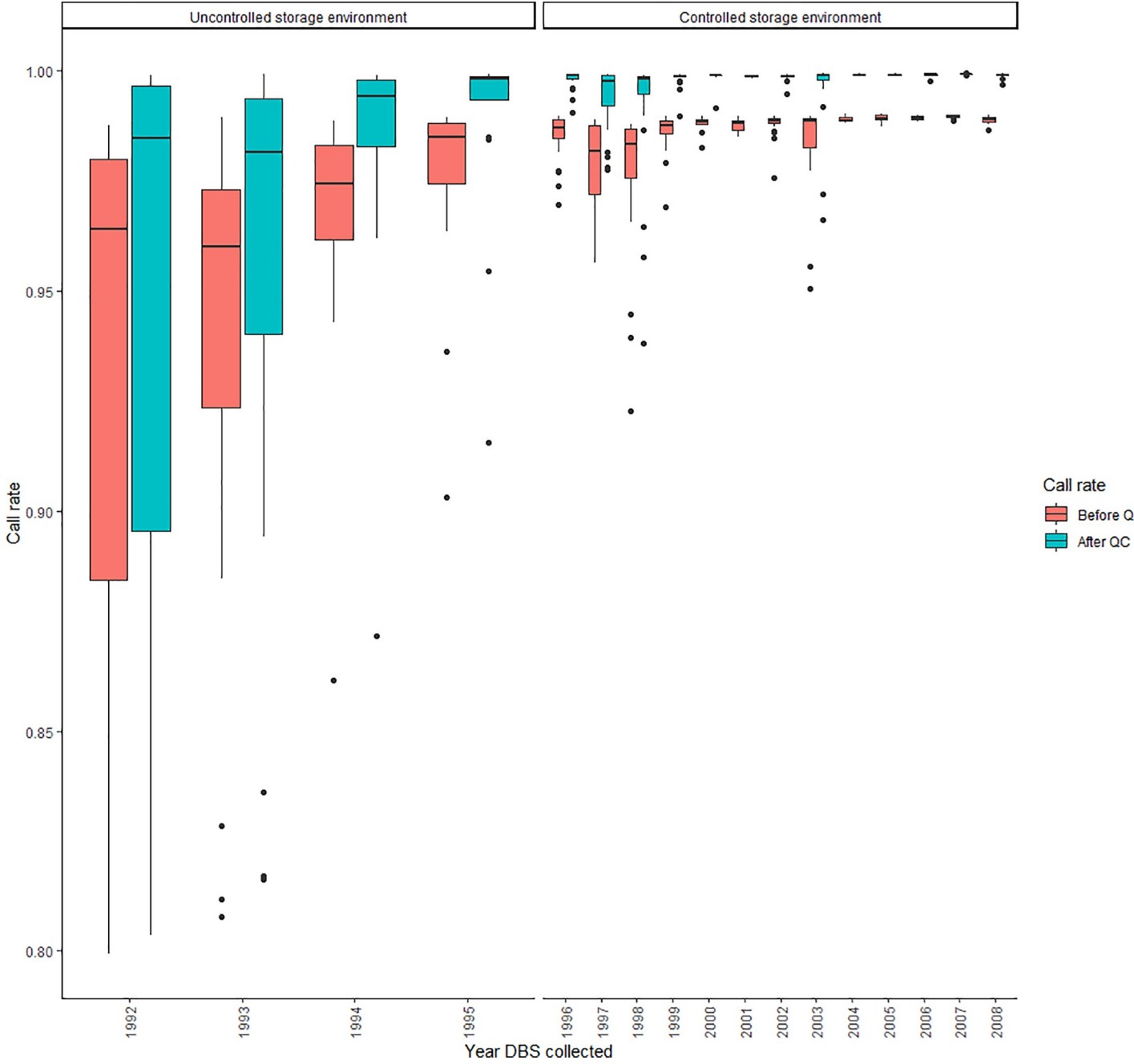

**Fig 1. Before and after variant filtering call rate of DBS archived at the Michigan Neonatal Biobank between 1992 and 2008.**

for 2.17% of all variants evaluated. The frequency of differential missingness was similar across each chromosome (range: 1.33–2.77%), suggesting storage condition adversely impaired DNA quality for genotyping indiscriminately.

## Discussion

Utilizing archived residual DBS for genome-wide association research has increased in recent years [24, 25]. DBS have become a rich resource for population-based epidemiologic research

**Table 2. Trend in proportions of call rate by collection years among DBS archived at the Michigan Neonatal Biobank between 1992 and 2008.**

| Storage environments | Year DBS collected | Total DBS collected | DBS proportion with call rate ≥ 99.0% | p-trend[1] |
|---|---|---|---|---|
| Uncontrolled environment[2] | | | | 0.006 |
| | 1992 | 26 | 0.42 | |
| | 1993 | 24 | 0.33 | |
| | 1994 | 29 | 0.59 | |
| | 1995 | 22 | 0.82 | |
| Controlled environment[3] | 1996 | 26 | 1.00 | 0.002 |
| | 1997 | 24 | 0.79 | |
| | 1998 | 24 | 0.79 | |
| | 1999 | 32 | 0.97 | |
| | 2000 | 16 | 1.00 | |
| | 2001 | 25 | 1.00 | |
| | 2002 | 28 | 1.00 | |
| | 2003 | 17 | 0.88 | |
| | 2004 | 8 | 1.00 | |
| | 2005 | 24 | 1.00 | |
| | 2006 | 33 | 1.00 | |
| | 2007 | 14 | 1.00 | |
| | 2008 | 27 | 1.00 | |

[1]Multivariable logistic regression model adjusted for DNA yield and maternal race/ethnicity.

[2]Uncontrolled temperature, 35% humidity.

[3]70 degrees Fahrenheit, 35% humidity.

on genetic factors of numerous diseases because of the accumulation of stored DBS along with the possibility to link samples to demographic and clinical data. However, due to the variability in the condition and environment that DBS are collected and archived within and between biobanks, it is good practice to consider the storage conditions and duration to evaluate the quality of extracted DNA across samples being used for any genomic studies. We evaluated sample call rate, a measurement of genotyping quality, of DBS archived at the MNB under varying storage conditions since their collection at births between 1992 and 2008. While controlled storage environment and newer blood spots yielded higher call rates, overall, our findings suggest DBS from MNB are viable and a potentially valuable resource for genome-wide association studies of common genetic variants.

In our analysis, storage environment and duration were the only two factors associated with differences in call rate. Specifically, DBS archived in a temperature- and humidity-controlled environment yielded higher DNA call rates (median call rate = 99.9%) when compared to those stored in an uncontrolled environment (median call rate = 99.3%). This is consistent with previous reports suggesting DBS storage environment influences biomarker stability [26, 27]. However, it should be noted that the average call rate for samples stored in an uncontrolled environment was acceptable for most genotyping applications. In this study, we used a call rate threshold of ≥ 99.0% to identify "high-quality" genotypes. Based on this, 85% of the selected samples were considered high quality (96% for those archived in a controlled environment). Moreover, many genome-wide association studies apply a less stringent call rate threshold of ≥ 95.0% to identify samples of acceptable quality [20, 21]. Using that threshold, 95.5% of the total DBS had sample call rates exceeding 95.0%, including 83% of the samples stored in an uncontrolled environment. Therefore, the majority of DBS regardless of their archival condition and duration should pass conventional quality control filtering applied in most

**Table 3. Proportions of differential missing SNPs by storage environment among DBS archived at the Michigan Neonatal Biobank between 1992 and 2008.**

| Chromosome | Number of SNPs evaluated | Number of SNPs with p-value $\leq 5\times10^{-8}$ | Proportion of SNPs with p-value $\leq 5\times10^{-8}$ |
|---|---|---|---|
| 1 | 26966 | 521 | 0.019 |
| 2 | 28580 | 630 | 0.022 |
| 3 | 24467 | 552 | 0.023 |
| 4 | 23160 | 641 | 0.028 |
| 5 | 21205 | 512 | 0.024 |
| 6 | 26951 | 628 | 0.023 |
| 7 | 20210 | 495 | 0.024 |
| 8 | 18607 | 461 | 0.025 |
| 9 | 15502 | 328 | 0.021 |
| 10 | 17538 | 355 | 0.020 |
| 11 | 17105 | 331 | 0.019 |
| 12 | 16769 | 352 | 0.021 |
| 13 | 12769 | 353 | 0.028 |
| 14 | 11316 | 252 | 0.022 |
| 15 | 10529 | 188 | 0.018 |
| 16 | 11242 | 164 | 0.015 |
| 17 | 10895 | 187 | 0.017 |
| 18 | 10504 | 247 | 0.024 |
| 19 | 7836 | 121 | 0.015 |
| 20 | 8521 | 120 | 0.014 |
| 22 | 5121 | 68 | 0.013 |
| All chromosomes | 345793 | 7506 | 0.022 |

contemporary genetic association studies. While proper archival of residual DBS in controlled environments will likely lessen DNA degradation over time, for the purposes of genotyping on array-based technology, DBS stored in an uncontrolled environment are still useable for most genetic studies. Additionally, our results suggest that DNA quality was consistent across the genome regardless of storage environments, indicating that little if any bias exists in evaluating certain genomic regions.

As noted, storage duration affected call rates regardless of DBS storage environment, where the most recently archived DBS yielded higher DNA call rates. This association was similarly described previously [28, 29], with the effect of storage duration on DNA quality reduced among DBS archived in a controlled environment. However, despite the significant association between storage duration and call rate, the vast majority of DBS archived in a controlled environment had a call rate $\geq 99.0\%$ (the exceptions are collection year 1997, 1998, and 2003 where there were DBS with call rate $< 99.0\%$). Conversely, among DBS archived in an uncontrolled environment, the proportion of DBS with call rate $\geq 99.0\%$ seems to increase linearly by storage year (Table 2), suggesting an improved call rate in more recently collected DBS.

This study evaluated the impact of environmental condition, storage duration, and demographic factors on genotyping quality obtained from nearly 400 residual DBS archived over two decades. While this study provides important information on correlates of DBS sample quality for array-based genotyping, the study findings should be considered in light of some limitations. First, our analyses are limited to assessment of DNA used for genotyping studies. More specifically, it is not clear if the quality of DNA extracted from DBS is sufficient for the detection of variants using whole-exome or whole-genome sequencing technologies. Notably, we obtained relatively high DNA yields from DBS [30, 31]; however, total yield may not

accurately reflect DNA quality and DNA degradation remains a concern. For more accurate estimations of quality DNA yield, future studies using DBS may want to consider specifically measuring dsDNA, which was not done in the current study. Additionally, because of our limited sample size, we were not able to fully evaluate the quality of low frequency variants (i.e., <1%) that are sometimes included in genome-wide association studies. Finally, because the samples included in this study were collected from a single bloodspot biobank, our findings may not be generalizable to DBS obtained from other biobanks with incomparable DBS archival environments.

In summary, residual DBS are a unique resource that facilitate population-based epidemiologic investigations of inherited genetic variation, epigenetic profiles, metabolomics and other potential biomarkers of complex diseases [27, 32–34], especially for rare pediatric outcomes. The results of this study illustrate that residual DBS archived at MNB provide high quality DNA for genetic association studies evaluating common variation using standard array-based genotyping. While we identified differences in average sample quality across storage environment and storage duration, these factors did not substantially limit the number of samples that would pass conventional quality control measures for most genome-wide association studies. For investigations of common phenotypes or limited sample sizes, priority may be given to more recently collected DBS and those stored in controlled environments. However, our findings generally support the use of DBS stored for extended periods in uncontrolled environments for investigations of rare phenotypes, which may require accruing cases occurring over a period of several decades in order to achieve acceptable sample sizes.

## Supporting information

**S1 Table. Associations between residual DBS characteristics and storage environment among DBS archived at the Michigan Neonatal Biobank between 1992 and 2008.**
(DOCX)

**S2 Table. Counts of residual DBS call rate, storage environment, and collection period by genotyping array position among DBS archived at the Michigan Neonatal Biobank between 1992 and 2008.**
(DOCX)

**S1 Fig. Manhattan plot showing the association of SNPs with storage environment among DBS archived at the Michigan Neonatal Biobank between 1992 and 2008.** Black and red horizontal lines signify p-value $\leq 1 \times 10^{-5}$ and $\leq 5 \times 10^{-8}$, respectively.
(TIFF)

## Acknowledgments

We want to thank the Michigan Department of Health and Human Services and the Michigan Neonatal Biobank for their input and feedback on the acquisition and use of residual newborn screening dried blood spot specimens.

## Author Contributions

**Conceptualization:** Philip J. Lupo, Michael E. Scheurer, Austin L. Brown.

**Data curation:** Erik A. Ehli, Noah A. Kallsen, Gareth E. Davies, Austin L. Brown.

**Formal analysis:** Pagna Sok.

**Funding acquisition:** Philip J. Lupo, Karen R. Rabin, Austin L. Brown.

**Investigation:** Karen R. Rabin.

**Methodology:** Philip J. Lupo, Melissa A. Richard, Erik A. Ehli, Noah A. Kallsen, Gareth E. Davies, Austin L. Brown.

**Supervision:** Philip J. Lupo, Michael E. Scheurer.

**Validation:** Pagna Sok.

**Visualization:** Pagna Sok.

**Writing – original draft:** Pagna Sok.

**Writing – review & editing:** Pagna Sok, Philip J. Lupo, Melissa A. Richard, Karen R. Rabin, Erik A. Ehli, Noah A. Kallsen, Gareth E. Davies, Michael E. Scheurer, Austin L. Brown.

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
