## [Decision Letter · Decision Letter 0]

12 Dec 2019

PONE-D-19-32165

Utilization of archived neonatal dried blood spots for genome-wide genotyping

PLOS ONE

Dear Dr Sok,

Thank you for submitting your manuscript to PLOS ONE. After careful consideration, we feel that it has merit but does not fully meet PLOS ONE’s publication criteria as it currently stands. Therefore, we invite you to submit a revised version of the manuscript that addresses the minor points raised during the review process.

We would appreciate receiving your revised manuscript by 6 weeks. To enhance the reproducibility of your results, we recommend that if applicable you deposit your laboratory protocols in protocols.io, where a protocol can be assigned its own identifier (DOI) such that it can be cited independently in the future. For instructions see: http://journals.plos.org/plosone/s/submission-guidelines#loc-laboratory-protocols

We look forward to receiving your revised manuscript.

Kind regards,

Isabelle Chemin, PhD

Academic Editor

PLOS ONE

Journal Requirements:

Reviewers' comments:

Reviewer's Responses to Questions

**Comments to the Author**

1. Is the manuscript technically sound, and do the data support the conclusions?

Reviewer #1: Yes

2. Has the statistical analysis been performed appropriately and rigorously? 

Reviewer #1: Yes

3. Have the authors made all data underlying the findings in their manuscript fully available?

Reviewer #1: No

4. Is the manuscript presented in an intelligible fashion and written in standard English?

Reviewer #1: Yes

5. Review Comments to the Author

Reviewer #1: The paper by Sok and colleagues explores how storage of dried blot spots cards may affect the final data and which factors that could compromise results. The manuscript is well written and easy to read.

The Introduction is overall a good review of the literature.

• I disagree that 13-15 are good references for GWAS, I would characterize these as methodological papers not unlike the one in review here. None-the-less the point is valid, there are numerous examples of canonical GWAS’s done on DBS, for example out of the iPSYCH/PGC.

The methodology is appropriate there are many alternative methods and here they have chosen one specific for the filter paper they use. Material on a DBS card is scarce which can be adjusted for by using more spots, here six are used. The statistics are appropriate and the authors are wise to exclude chromosome 21 since, as stated, the methodology does poorly when deviating from chr_n=2. The results are appropriately presented.

• You found no significance between in concentrations between +/- controlled, what about between concentration and call rate? Could the difference in CR>99% be confounded by low concentration?

• Also the yields are high i.e. µg. https://www.ncbi.nlm.nih.gov/pmc/articles/PMC1735924/ estimated theoretical yields, recalculating this for a 3mm spot you get 60ng as done in https://www.ncbi.nlm.nih.gov/pubmed/19575812. here your median estimate is almost a order of magnitude higher. Did you consider messuring concentrations with a dsDNA specific assay?

• Probes were excluded if the heterozygosity rate exceeds 0.4/40%, I am puzzled by this and curious as to why? For a variant with a MAF=0.5 the expected equilibrium would be 25%AA, 25%BB and 50%AB, thus such variant could exist in a natural outbreeding population meaning you exclude valid genotypes. The loss for this is high removing ~15% of the array content as per the authors own numbers (100K/700K).

• The physical location on the chip has been known to affect call rate, presumably driven by concentration gradients over the stain flow chambers. Especially the “top” of the chip has been prone to failure on the GSA v1. Did you consider chip placement as a variable?

• For the same reason above I am slightly skeptical about the trends in table 2. Only 3:13 point deviate from absolute success (prop>99%=1) meaning those three are sure to act as leverage points. Could this just be driven by experimental artifacts from samples placement on underperforming array?

I find this to be an interesting paper with few important messages. First is that when storing samples a controlled environment is preferable, second being that even in an uncontrolled environment the genotyping by array approach generates robust and reliable results despite decades of storage.

A general weakness of these papers is that there is no consensus in neonatal biobanks on how to store samples. As done here, showing that even the uncontrolled samples have great value is important. These population samples have many advantages over more typical samples of convenience.

6. PLOS authors have the option to publish the peer review history of their article (what does this mean?). If published, this will include your full peer review and any attached files.

Reviewer #1: No

---

## [Author Response · Author response to Decision Letter 0]

8 Jan 2020

Dear editor:

Thank you for the opportunity to revise and resubmit our manuscript, entitled “Utilization of archived neonatal dried blood spots for genome-wide genotyping.” We found all of the reviewer comments to be thoughtful and well-informed. We carefully considered each suggestion as we revised the manuscript, and we believe that addressing the reviewer’s concerns strengthened the manuscript considerably. 

All revisions are indicated using track changes in the attached files. Additionally, we have included a second supplementary table (S2 Table) that addresses data concerning the associations between DBS call rate and genotyping array row position. The following are our responses and edits to the reviewers’ comments. 

Reviewers’ comment 1: I disagree that 13-15 are good references for GWAS, I would characterize these as methodological papers not unlike the one in review here. None-the-less the point is valid, there are numerous examples of canonical GWAS’s done on DBS, for example out of the iPSYCH/PGC.

The methodology is appropriate there are many alternative methods and here they have chosen one specific for the filter paper they use. Material on a DBS card is scarce which can be adjusted for by using more spots, here six are used. The statistics are appropriate and the authors are wise to exclude chromosome 21 since, as stated, the methodology does poorly when deviating from chr_n=2. The results are appropriately presented.

Authors’ response 1: We appreciate the reviewers careful and thoughtful consideration of the manuscript. We agree that the references in question were not the most appropriate for GWAS of DBS. We have moved these refences to more accurately reflect the content of these papers and included new references to provide examples of GWAS using DBS. 

Reviewers’ comment 2: You found no significance between in concentrations between +/- controlled, what about between concentration and call rate? Could the difference in CR>99% be confounded by low concentration?

Authors’ response 2: As correctly indicated by the reviewer, we did not identify a statistically significant difference in total DNA yield between storage environments (p=0.05), as shown in supplementary table 1. As requested, we have included a comparison of total DNA yield by call rate in Table 1. Again, we did not observe a significant association (p=0.3). Given the relatively weak associations observed, we do not believe total DNA yield is a strong confounder of the association between storage environment and call rate. Notably, including DNA yield as a covariate in our multivariable regression models did not materially impact our results.

Reviewers’ comment 3: Also the yields are high i.e. µg. https://www.ncbi.nlm.nih.gov/pmc/articles/PMC1735924/ estimated theoretical yields, recalculating this for a 3mm spot you get 60ng as done in https://www.ncbi.nlm.nih.gov/pubmed/19575812. here your median estimate is almost a order of magnitude higher. Did you consider measuring concentrations with a dsDNA specific assay?

Authors’ response 3: The reviewer raises an important limitation of our study. Although the yields were quite high in the current study, we did not specifically assess dsDNA concentrations. We have noted this limitation in our discussion section.

Reviewer’s comment 4: Probes were excluded if the heterozygosity rate exceeds 0.4/40%, I am puzzled by this and curious as to why? For a variant with a MAF=0.5 the expected equilibrium would be 25%AA, 25%BB and 50%AB, thus such variant could exist in a natural outbreeding population meaning you exclude valid genotypes. The loss for this is high removing ~15% of the array content as per the authors own numbers (100K/700K).

Authors’ response 4: Although the heterozygosity rate of 0.4/40% exclusion threshold is suggested by GenomeStudio clustering algorithm metrics, indicated in table 1 of the Infinium Genotyping Data Analysis guide (link included below), we agree with the reviewer’s concern. This exclusion criterion may be overly restrictive. Therefore, we revised our quality control measures in the DNA genotyping and quality control section of the Method (line 98-111 of the manuscript). Updated call rates were calculated based on the 472,019 SNPs that passed quality control. We then re-evaluated our analyses using the new call rates. All updated findings are indicated through track changes. Overall, our updated results are consistent with our initial findings.

https://www.illumina.com/Documents/products/technotes/technote_infinium_genotyping_data_analysis.pdf

Reviewers’ comments 5: The physical location on the chip has been known to affect call rate, presumably driven by concentration gradients over the stain flow chambers. Especially the “top” of the chip has been prone to failure on the GSA v1. Did you consider chip placement as a variable?

Authors’ response 5: Thank you for raising this important point. Notably, chip position was randomly assigned to all samples included in this study, a detail we have added to our results section. Still, we agreed with the reviewers that physical location on the genotyping array can affect sample call rate. Thus, we decided to include genotyping array position as a potential factor affecting sample call rate. With row 12 indicating the top of the array and row 1 the bottom, we tabulated sample call rate and storage environment by genotyping array row position. All results were reported in the supplemental table 2; however, we did not observe significant association for call rate (p=0.92) or storage environment (p=0.99) by array positioning. 

Reviewers’ comments 6: For the same reason above I am slightly skeptical about the trends in table 2. Only 3:13 point deviate from absolute success (prop>99%=1) meaning those three are sure to act as leverage points. Could this just be driven by experimental artifacts from samples placement on underperforming array?

Authors’ response 6: The reviewer raises a valid concern. As reported in the supplementary table 2, we did not observe significant association between storage environment (p=0.99) or DBS collection period (p=0.99) and array positioning. Hence, we do not believe the significant p-trends in table 2 is confounded by DBS position on the genotyping array.

---

## [Decision Letter · Decision Letter 1]

5 Feb 2020

Utilization of archived neonatal dried blood spots for genome-wide genotyping

PONE-D-19-32165R1

Dear Dr. Sok,

We are pleased to inform you that your manuscript has been judged scientifically suitable for publication and will be formally accepted for publication once it complies with all outstanding technical requirements.

With kind regards,

Isabelle Chemin, PhD

Academic Editor

PLOS ONE

Additional Editor Comments (optional):

Reviewers' comments:

Reviewer's Responses to Questions

**Comments to the Author**

1. If the authors have adequately addressed your comments raised in a previous round of review and you feel that this manuscript is now acceptable for publication, you may indicate that here to bypass the “Comments to the Author” section, enter your conflict of interest statement in the “Confidential to Editor” section, and submit your "Accept" recommendation.

Reviewer #1: All comments have been addressed

2. Is the manuscript technically sound, and do the data support the conclusions?

Reviewer #1: Yes

3. Has the statistical analysis been performed appropriately and rigorously? 

Reviewer #1: Yes

4. Have the authors made all data underlying the findings in their manuscript fully available?

Reviewer #1: Yes

5. Is the manuscript presented in an intelligible fashion and written in standard English?

Reviewer #1: Yes

6. Review Comments to the Author

Reviewer #1: Very good response to the questions/concerns raised during review.

In my opinion, this is interesting reading for anyone who considers using this type of material.

7. PLOS authors have the option to publish the peer review history of their article (what does this mean?). If published, this will include your full peer review and any attached files.

Reviewer #1: Yes: Jonas Bybjerg-Grauholm

---

## [Editor Report · Acceptance letter]

7 Feb 2020

PONE-D-19-32165R1 

Utilization of archived neonatal dried blood spots for genome-wide genotyping 

Dear Dr. Sok:

I am pleased to inform you that your manuscript has been deemed suitable for publication in PLOS ONE. Congratulations! Your manuscript is now with our production department. 

With kind regards,

on behalf of

Mrs Isabelle Chemin 

Academic Editor

PLOS ONE